# Evaluation of the *forAge* Age-at-Death Estimation Program Using Pubic Symphyseal Surface in a Korean Population

**DOI:** 10.3390/diagnostics14080793

**Published:** 2024-04-10

**Authors:** Hyun Jin Park, Sehyun Song, Eun Jin Woo, Kyung-Seok Hu

**Affiliations:** 1Department of Anatomy, Daegu Catholic University School of Medicine, Daegu 42472, Republic of Korea; hjpark321@cu.ac.kr; 2Department of Oral Biology, Division in Anatomy and Developmental Biology, Human Identification Research Institute, BK21 FOUR Project, Yonsei University College of Dentistry, Seoul 03722, Republic of Korea; bokcool21@gmail.com; 3Department of History, Sejong University, Seoul 05006, Republic of Korea

**Keywords:** pubic symphyseal surface, 3-dimensional analysis, age-at-death estimation, *forAge*, Korean

## Abstract

The *forAge* program estimates the age-at-death of human pubic symphysis using 3-dimensional scans. It was developed by Dennis E. Slice and Bridget F. B. Algee-Hewitt, and utilizes three distinct scores: the Slice and Algee-Hewitt (SAH) score, bending energy (BE), and ventral curvature (VC). However, these scores and age estimation regression equation were obtained through European American pubic symphysis. Changes in the pubic symphysis surface are evaluated as one of the most reliable indicators for estimating age, but in connection with this, using Korean materials, changes in the pubic symphysis surface and the actual changes are evaluated. There is no bar where the relationship between ages is grasped, and there are cases where a methodology developed for a specific group is applied to a Korean group. Changing the pubic symphysis surface by aging was evaluated as one of the most reliable indicators for estimating age. However, there is no study conducted on the relationship between changes in the pubic symphysis and actual age and applied the age estimation method for a specific population among Korean population. The purpose of this study is to compare the difference between the actual age and the estimated age in Korean to see if the *forAge* program is applicable to other population of different ancestral origin. One hundred and four modern Korean pubic symphyseal surfaces (47 to 96 years old) were used in this study. Through the pubic symphyseal surface 3-dimensional images, age-at-death was estimated via prediction equation and new regression lines using SAH, VC, and BE scores. Firstly, the estimated age via prediction equation using the first version of SAH score was lower than the actual age according to all pubic symphyseal surfaces for those older than 56. With aging, the difference between the actual age and estimated age became markedly larger. Secondly, the estimated ages via the new regression lines using VC, the second version of SAH score, and BE were shown a similar pattern to the previous prediction equation. The current study explored the applicability of a quantitative method using pubic symphyseal surface for age estimation in a modern Korean population. This study showed the *forAge* program cannot be applied to a modern Korean population, as they present relatively low correlations with the actual age-at-death.

## 1. Introduction

Forensic anthropologists use skeletal markers related to bone resorption, deposition, and degeneration to estimate an individual’s age. Various parts of the skeleton, including cranial [1] and palatine [2] sutures, clavicle [3], rib [4], pubic symphysis [5], iliac auricular surface [6], acetabulum [7], and sacrum [8], have been employed in age estimation. Among the many skeletal indicators, pubic symphyseal surface, iliac auricular surface, acetabulum, sternal rib ends, and sternal end of the clavicle have been deemed the most reliable skeletal age indicators for an aging adult [9].

Recently, studies focusing on sex determination using specific parts of bones such as the tibia, fibula, and various parts of the skull are also being conducted in Korea [10,11]. Additionally, studies estimating stature using long bones such as the upper limb, lower limb, vertebrae, and sacrum have been increasingly conducted in the Korean population [12]. However, research in the field of age estimation utilizing Korean skeletal remains has been quite limited, often confined to histological analysis of tibia and fibula tissues, morphometric analysis of thyroid cartilage calcification, tooth attrition, or age-group comparison through bone density measurements [13,14,15].

Over the last several decades, age determination using the pubic symphyseal surface (of pubic symphysis) has been extensively researched [5,16,17,18,19,20]. In young adults, the pubic symphyseal surface is irregular and billowy, but it becomes smoother and shows degenerative change throughout adulthood and senescence [5,21]. Therefore, it serves as a representative skeletal site to estimate the age-at-death of humans. The Suchey–Brooks system [5], which excludes ancestral origins, is considered the gold standard for pubic symphyseal surface age determination.

The *forAge* program is a software that estimates the age-at-death of the human pubic symphyseal surface based on 3-dimensional (3D) scans [22,23]. It was devised based on the Suchey–Brooks system, in which the pubic symphyseal surface becomes flat, and the ventral edge changes from an oval to a circle with aging. For a multivariate approach, the program was designed using the SAH method, VC, and BE. Using the aforementioned methods, the *forAge* program calculates the estimated age-at-death [23].

Recently, the *forAge* age estimation program was evaluated in multiple ancestral origins. In European males [24], for the samples under 40 years, the root-mean-square-error (RMSE) was 5.93 to 7.48 years, except for the method using VC score. However, individuals up to 40 years, RMSE and inaccuracy were sharply increased from 22.82 to 28.88 and 19.42 to 26.04 years, respectively. In White South Africans [25], the *forAge* underestimated age-at-death after the sample age of 35 years, and also this age estimating method performed better in male than female sample.

In the realm of forensic anthropology, the precise estimation of age-at-death stands as a cornerstone, pivotal for the identification of remains and the unravelling of historical narratives. The *forAge* program emerges as a beacon in this scientific domain, heralding an era of enhanced precision and nuanced understanding in age estimation methodologies. This innovative program is not merely an analytical tool; it represents the confluence of advanced statistical techniques and a profound comprehension of human osteology.

The evolution of age-at-death estimation methods has been marked by a transition from rudimentary observations to sophisticated, data-driven approaches. The *forAge* program epitomizes this evolution, encapsulating a rich database of demographic variability and anatomical detail. Its algorithmic precision reflects a deep integration of biological realities with statistical rigor, offering a lens through which the subtleties of skeletal aging are not just observed, but intricately understood.

This study delves into the *forAge* program’s capabilities, aiming to underscore its potential not just as a forensic tool, but as a catalyst in the broader scientific quest to comprehend the human lifespan. By examining its application in diverse scenarios and comparing its outputs against established benchmarks, the study seeks to illuminate the program’s strengths and explore its role in shaping the future trajectory of forensic anthropology.

However, despite these global advancements and the increasing focus on sex determination and stature estimation using skeletal remains, it is noteworthy that such research, specifically in the field of age estimation utilizing skeletal remains, has not been extensively pursued in Korea. This lack of localized research underscores a significant gap in the forensic anthropology literature, and points to the need for more targeted studies that consider the unique characteristics of the Korean population.

This study seeks to address this gap by exploring the capabilities of the *forAge* program. By examining its application in diverse scenarios and comparing its outputs against established benchmarks, the study aims to highlight the program’s strengths and delineate its role in advancing forensic anthropology, particularly within the Korean context.

Thus, the methods and age estimation regression equations of the *forAge* program showed different results for each ancestral origin. Therefore, it would be meaningful to compare the difference between the actual age and the estimated age for adult age-at-death in a modern Korean population using this program.

## 2. Materials and Methods

### 2.1. Sample Properties

To evaluate the efficacy and accuracy of the *forAge* program specifically within the Korean population, 104 sides of Korean pubic symphyseal surfaces, donated to the Yonsei University College of Medicine, were utilized. These specimens included 39 males and 19 females, aged between 47 to 96 years, with a mean age of 74.5 years, highlighting a focus on middle-aged to elderly demographics. The pubic symphyseal surfaces showing damage from either dissection or bone maceration processes were rigorously excluded. Consequently, the final distribution of the pubic symphyseal surface samples analyzed in the study was comprised of 10 from specimens in the 40 to 50 age group, 19 from those in their 60s, 38 from those in their 70s, 33 from those in their 80s, and 4 from those in their 90s (Figure 1).

### 2.2. Bone Preparation

None of the cadaveric specimens had congenital malformations, pathological findings, and a history of surgery or trauma. The study was conducted in accordance with the Declaration of Helsinki ethical principles for medical research involving human subjects. All authors were well-informed of the WMA Declaration of Helsinki—Ethical Principles for Medical Research Involving Human Subjects—and confirmed that the present study firmly fulfilled the declaration. The cadaveric specimens were legally donated to Yonsei University College of Medicine.

The preparation of the pubic bones commenced with manual dissection, removing the majority of soft tissue to expose the skeletal features critical for subsequent analysis. This was followed by a simmering process, in which the bones were immersed in a 90 °C distilled water bath containing a 1% Yuhangen^®^ (Yuhan Corporation, Seoul, Republic of Korea) for 72 h. Subsequent to simmering, any remaining soft tissues adhering to the pubic symphyseal surface were carefully removed with a periosteal elevator to ensure minimal interference with the bone surface. Afterwards, the bones were then allowed to dry at room temperature for 24 h, followed by immersion in 99.9% ethyl alcohol for 48 h to remove any residual water.

### 2.3. Data Acquisition and Analysis

Regardless of the different sides, the pubic symphyseal surface was scanned using an Artec Space Spider scanner (Artec 3D, Luxembourg, Luxembourg), which is a high-resolution 3D scanner based on blue light technology, with a 3D resolution of 0.1 mm and a 3D point accuracy of 0.05 mm. In the scanning phase, each specimen was precisely positioned on a rotary table, allowing for a uniform and detailed capture of the pubic symphyseal surface in a single rotation. This procedure ensured high-fidelity recording of essential anatomical features, minimizing the risk of data loss or geometric distortion. After scanning, the raw data was rigorously processed using Artec Studio 15 software (Artec 3D). During this stage, noise and artifacts were meticulously removed, a crucial step to enhance the quality of the scans for accurate age estimation. The processed 3D scans were then carefully trimmed to isolate the pubic symphyseal surface, using Autodesk ReCap Photo (Autodesk^®^, San Rafael, CA, USA). The resulting files were exported in the Polygon File Format (.ply) and precisely prepared to satisfy the input requirements of the *forAge* program.

The *forAge* program, an open-source software (Ver. 20180217) developed by the Department of Scientific Computing at Florida State University, was employed for the final analysis. Accessible at http://morphlab.sc.fsu.edu/software/forAge/index.html (accessed on 1 August 2022), this program is at the forefront of age estimation technology.

It was utilized to process the prepared .ply files, executing complex computations to derive age estimations. The program’s algorithm is designed to calculate five distinct point estimates of age for each individual, including the Slice and Algee-Hewitt (SAH) score, bending energy (BE), and ventral curvature (VC) estimates. Furthermore, it combines these metrics—VC and SAH, and VC and BE—to provide a multifaceted and robust analysis, harnessing the full potential of the data captured from the pubic symphyseal surface.

## 3. Results

### 3.1. Evaluation of Prediction Equation Using the SAH (Ver. 1) Score

The results showed that the prediction equation, which utilizes the SAH (Version 1) score [22] for age estimation within the Korean population, revealed clear patterns (Figure 2). For individuals older than 56 years, the age estimated by the equation was consistently lower than their actual age across all examined pubic symphyseal surfaces. Notably, for those younger than 51 years, the estimated age was consistently higher than the actual age. The root-mean-squared-error (RMSE), indicative of the model’s prediction accuracy, showed an increasing trend with age: 7.37 in the 40 to 50s age group, 7.33 in the 60s, 17.35 in the 70s, 26.60 in the 80s, and 34.78 in the 90s. This trend suggests an increasing divergence between the actual and estimated ages as age progresses.

When k = 2 was applied in the prediction model, the estimated age range accurately encompassed individuals up to their 60s. However, notable deviations began to emerge for those in their 70s. While the early 70s were still within the prediction age range, the accuracy significantly decreased for individuals in the later years of this age group, with the predicted age range failing to encompass the actual ages of these older individuals. This discrepancy was even more evident for individuals in their 80s, where all but three samples exceeded the predicted age range. In the case of individuals in their 90s, the actual ages were found to be over 10 years greater than the maximum value of the predicted age range, indicating a significant underestimation by the model for the oldest age group (Figure 2). This pattern of deviation highlights a critical limitation in the prediction model, particularly in its application to the elderly population, suggesting the need for further refinement for more accurate age estimation in this age range.

### 3.2. Evaluation of New Regression Lines Using the VC, SAH (Ver. 2), BE Score

Second, in the comprehensive analysis of the new regression lines using VC, SAH (Version 2), and BE scores [23], the study consistently observed an underestimation of actual age across all pubic symphyseal surfaces examined. This trend was evident when analyzing the performance of each regression line individually. Particularly, the regression line based on the VC score stood out for its precision, as evidenced by the comparatively lower RMSE values, suggesting a higher accuracy in age estimation within this model.

More specifically, the RMSE values associated with the VC-based regression line delineated an increasing trajectory with advancing age groups. For individuals categorized in the younger age bracket, specifically those under 60 years, the RMSE was recorded at 15.72. This value exhibited a gradual rise across the successive age decades, marked at 19.83 for those in their 60s, reaching 32.50 in the 70s, 36.46 in the 80s, and peaking at 50.81 for individuals in their 90s. The detailed breakdown of these RMSE values can be found in Figure 3, providing a comprehensive view of the model’s performance across different age groups.

In a parallel comparative analysis focusing on the RMSE values across different regression lines, a distinctive pattern emerged. The regression line incorporating VC demonstrated the lowest RMSE, signifying this model as the most precise among those evaluated. The regression line with VC and SAH followed, displaying the next level of accuracy. Subsequently, the SAH model exhibited an increase in RMSE, and finally, the regression line combining VC and BE recorded the highest RMSE among the models considered. These comparative insights into the RMSE values, systematically tabulated in Table 1, offer a nuanced understanding of the varying levels of precision inherent in each regression line model.

This comprehensive dataset, emphasizing the trend of age underestimation and the comparative precision of different regression line models, provides a foundational platform for further discussion on the implications and potential avenues for refining age estimation techniques in forensic anthropology.

## 4. Discussion

The recent surge in computational capabilities and the development of advanced programming languages has revolutionized the field of forensic anthropology. These technological advancements have not only enabled more sophisticated analytical approaches, but have also significantly reduced the time required for method development and data processing [26]. The exploration of the pubic symphyseal surface through computational methods, as seen in recent studies, exemplifies this shift [27,28]. By employing enhanced computational techniques, researchers have been able to devise novel approaches for estimating age-at-death, offering alternatives to traditional phase-based methods which have certain limitations [22,23].

However, as with any emerging technique, thorough validation across various populations and conditions is imperative. This study’s focus on the Korean population is a vital part of this validation process, offering insights into how these advanced computational methods perform in a demographic that has not been extensively studied. Employing metrics such as inaccuracy and bias to evaluate these methods highlights their practical utility and potential limitations, ensuring their efficacy and reliability in diverse forensic applications.

Firstly, in the prediction equation using SAH, the first equation using the SAH score, all estimated ages were between 50 to 60 years in the Korean samples. This equation is the most accurate in 40- to 60-year samples (Figure 2A,B). The age of European Americans [22], differed in terms of inaccuracy value. Secondly, the results indicate significant differences in age estimation using SAH, BE, VC, VC and SAH, and VC and BE between the ancestral origin Figure 3). In European Americans, the bias and inaccuracy were around ±10 years for samples from the 50 to 59 age group, but in Koreans, the bias and inaccuracy were more than −25 to −11 years, even in the samples from the 40 to 50s age group. The bias was slightly higher for samples greater than 60 years than in the European American study, but the results were comparable. It was also confirmed in the samples from the 70s, 80s, and 90s age groups in this study, with inaccuracy values ranging from 31 to 67 (Table 2).

Although males outperformed females in White South African samples across all estimates in this program [25], there was no statistical significance between males and females in Korean samples. In this study, the age estimation model using VC, which estimated age-at-death through the curvature of the ventral margin of the pubic symphysis, showcased the lowest bias and inaccuracy values. In European males [24], RMSE, bias, and inaccuracy values were the lowest like in the Korean population. However, a weak correlation was observed in the White South African population (Table 3). It was found that the estimated age by the *forAge* program was different depending on ancestral origin.

Age estimation in adults presents a significant challenge for forensic anthropologists due to the complex nature of the aging process, individual variations, and the extensive range of environmental factors that affect it. Age provided by anthropologists is determined as a range rather than a specific value. The current study explored the applicability of a quantitative method using the pubic symphyseal surface for age estimation methods in a modern Korean population. Note that the age range determined for younger individuals was narrower than that for older individuals. Since this study used donated cadavers, the applicability of this method is limited to middle-aged and older Koreans. Thus, there is a need for further research to apply this type of quantitative method to a wider variety of age groups in Korea.

Moreover, this study underscores the importance of a multifaceted approach to age estimation in forensic anthropology. Integrating the *forAge* program with other forensic methodologies and databases could provide a more comprehensive and robust framework for age estimation, accommodating a broader spectrum of demographic variables and case-specific factors.

In conclusion, while the results did not align with the initial expectations of precision, they offer valuable insights into the *forAge* program’s current capabilities and limitations. This study sets a constructive path for future research, emphasizing the need for ongoing refinement and testing of the program to enhance its reliability and applicability in the field of forensic anthropology.

## 5. Conclusions

This study evaluated the applicability of the *forAge* age estimation program to Korean adults. This study determined that, in its current form, the *forAge* program could not be applied to the Korean population due to relatively low correlations with actual age and significantly high biases. Furthermore, there is no statistical relevance between male and female samples. The age-at-death estimation of pubic symphyseal surface for individuals older than 60 years leads to very inaccurate estimates with overly broad age intervals. In conclusion, a quantitative evaluation of age-related surface changes provides an objective assessment. However, future attempts to quantify surface changes should consider the biological capabilities of individual skeletal indicators to reflect age changes.

## Figures and Tables

**Figure 1 diagnostics-14-00793-f001:**
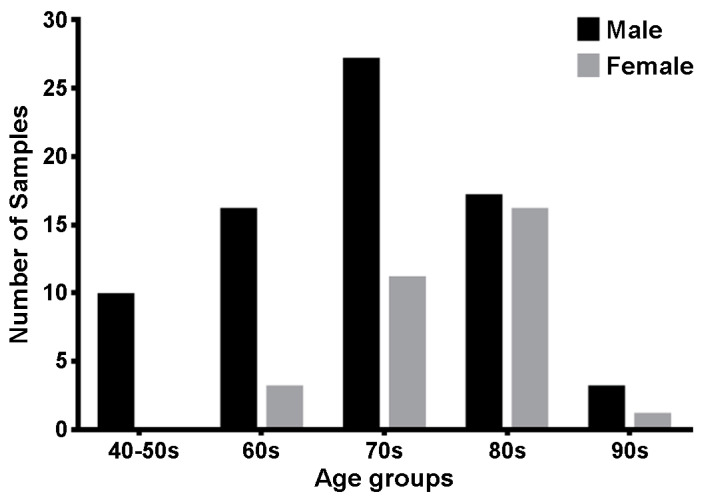
Distribution of number of samples according to age and sex. The number of pubic symphyseal surfaces was 10, 19, 38, 33, and 4 for those aged 40 to 59, 60 to 69, 70 to 79, 80 to 89, and 90 to 99, respectively. Because the samples in their 40s and 50s were too small, they were combined into a single group.

**Figure 2 diagnostics-14-00793-f002:**
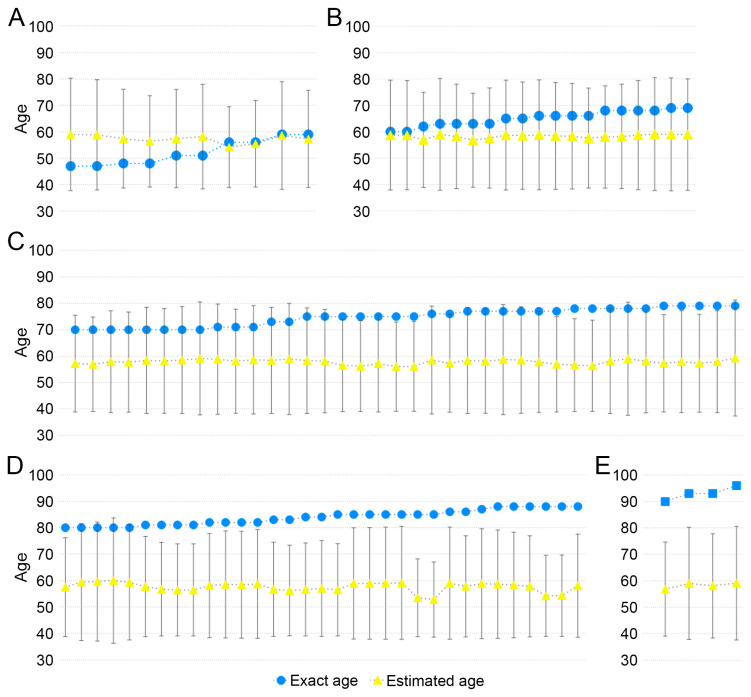
Exact age and estimated age using SAH score prediction equation. Exact age (blue circle) and estimate age (yellow triangle) via SAH score prediction equation of (**A**) 40 to 50s, (**B**) 60s, (**C**) 70s, (**D**) 80s, and (**E**) 90s samples. The error bars of the estimated age show the prediction age range (k = 2).

**Figure 3 diagnostics-14-00793-f003:**
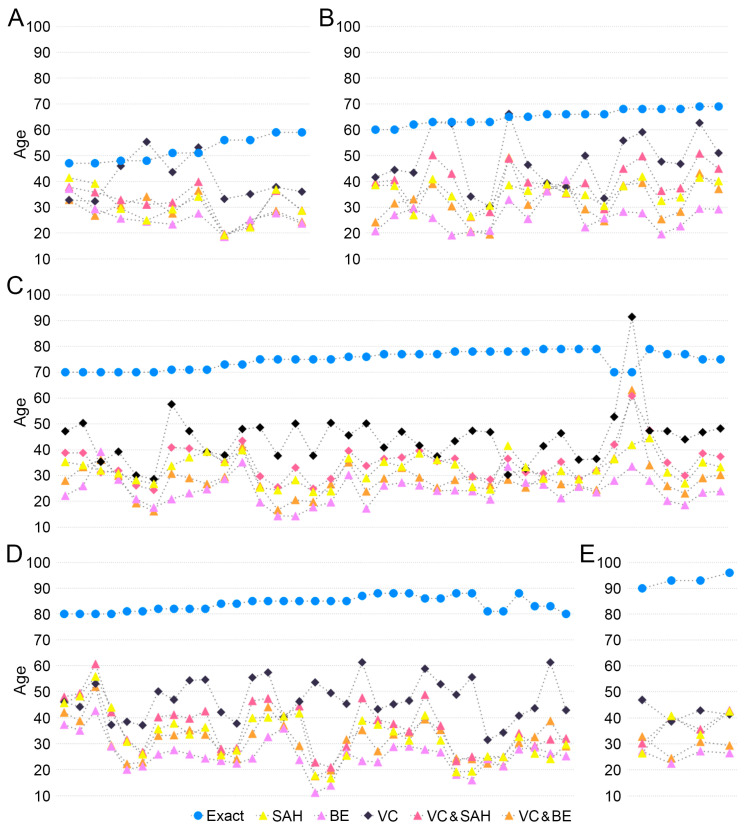
Results of the Korean samples applied to the *forAge* program. Exact age and estimated age using *forAge* for those in their (**A**) 40 to 50s, (**B**) 60s, (**C**) 70s, (**D**) 80s, and (**E**) 90s. Exact age (blue circle), estimated age using SAH (yellow triangle), BE (purple triangle), VC (black rhombus), VC and SAH (pink triangle), VC and BE (orange triangle).

**Table 1 diagnostics-14-00793-t001:** Root-mean-square-error of the new regression line [23].

	SAH	BE	VC	VC and SAH	VC and BE
40–50s	23.74	27.13	15.72	22.52	25.16
60s	29.77	39.07	19.83	26.49	33.31
70s	42.75	50.89	32.50	40.54	46.95
80s	52.02	58.64	37.46	48.68	53.18
90s	57.37	67.28	50.81	55.88	63.79

SAH, Slice and Algee-Hewitt; BE, bending energy; VC, ventral curvature.

**Table 2 diagnostics-14-00793-t002:** Bias and inaccuracy values of each age estimation models in the Korean samples.

	SAH (1st)		SAH (2nd)		BE		VC		VC & SAH		VC & BE	
Age-at-Death	Bias	Inaccuracy	Bias	Inaccuracy	Bias	Inaccuracy	Bias	Inaccuracy	Bias	Inaccuracy	Bias	Inaccuracy
40–50s	5.08	5.96	−21.68	21.68	−25.95	25.95	−11.68	13.57	−20.54	20.54	−23.68	23.68
60s	−6.83	6.83	−29.42	29.42	−38.66	38.66	−17.01	17.12	−25.54	25.54	−32.53	32.53
70s	−17.00	17.00	−42.24	42.24	−50.45	50.45	−30.44	31.57	−39.70	39.70	−46.07	46.07
80s	−26.35	26.35	−50.99	50.99	−58.14	58.14	−36.74	36.74	−47.55	47.55	−52.53	52.53
90s	−34.75	34.75	−57.19	57.19	−67.21	67.21	−50.60	50.60	−55.81	55.81	−63.64	63.64

SAH, Slice and Algee-Hewitt; BE, bending energy; VC, ventral curvature.

**Table 3 diagnostics-14-00793-t003:** Comparisons, in years, of RMSE, bias, and inaccuracy for each method. Results for the entire dataset of each population.

Regression Model		This Article	Joubert et al. [24]		Kotěrová et al. [25]
			Male	Female	
SAH (1st)	RMSE	19.90	-	-	-
	Bias	−16.67	−35.112	−41.792	-
	Inaccuracy	17.73	-	-	-
SAH (2nd)	RMSE	43.23	-	-	20.91
	Bias	−41.27	−12.785	−22.309	−13.58
	Inaccuracy	41.27	-	-	15.7
BE	RMSE	50.61	-	-	22.09
	Bias	−49.03	−25.033	−29.894	−15.54
	Inaccuracy	49.03	-	-	16.75
VC	RMSE	32.03	-	-	18.35
	Bias	−28.96	−16.713	−21.467	−8.43
	Inaccuracy	29.57	-	-	14.15
VC and SAH	RMSE	40.61	-	-	21.08
	Bias	−38.38	−12.765	−22.336	−13.67
	Inaccuracy	38.38	-	-	15.99
VC and BE	RMSE	46.08	-	-	22.25
	Bias	−44.17	−22.655	−28.474	−15.67
	Inaccuracy	44.17	-	-	16.96

SAH, Slice and Algee-Hewitt; BE, bending energy; VC, ventral curvature.

## Data Availability

The data presented in this study are available on request from the corresponding author.

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
