# Peer review of "Evaluation of the forAge Age-at-Death Estimation Program Using Pubic Symphyseal Surface in a Korean Population"

_diagnostics, 2024, doi:10.3390/diagnostics14080793_

Round 1

Reviewer 1 Report

Comments and Suggestions for Authors

The authors present an interesting study of age estimation based on symphysial pubic surfaces and its predictability through the use of the forAGE program on the Korean population. The overall quality of the manuscript is good. The manuscript is well-organized. The design of the study is well presented and explained. This study represents a useful contribution to the field.

Overall, the manuscript should undergo an English language revision, ideally by a native English speaker. Some typos should be corrected (e.g. energe should be replaced with energy).

Line 54: contains a reference that uses a different format from the other references in the manuscript: (12) instead of [12].

Author Response

We are grateful for the very constructive advice given by the reviewers. We did our best in revising the manuscript with available data and to share the excitement we had in this study.

Query 1. Overall, the manuscript should undergo an English language revision, ideally by a native English speaker. Some typos should be corrected (e.g. energe should be replaced with energy).

RESPONSE: Thank you for pointing out our oversight. The typo has been corrected, and the English has been reviewed for accuracy.

Query 2. Line 54: contains a reference that uses a different format from the other references in the manuscript: (12) instead of [12].

RESPONSE: Line 54 reference format was revised.

Reviewer 2 Report

Comments and Suggestions for Authors

I have read this paper with great interest and I think the topic is original. However, in the first subparagraph of Materials and Methods' section (line 113), authors state that the sample used consist of 104 pubic symphiseal surfaces, but then (line 115) they affirm "[...] The specimens, comprised of 39 males and 19 females [...]". Then, from line 118 to line 120, the sum of the subgroups of the sample again gives 104.

Line 161: there is an error in the last part of this line.

From line 167 to line 170: I think this part should be in the next paragraph.

Author Response

We are grateful for the very constructive advice given by the reviewers. We did our best in revising the manuscript with available data and to share the excitement we had in this study.

Query 1. I have read this paper with great interest and I think the topic is original. However, in the first subparagraph of Materials and Methods' section (line 113), authors state that the sample used consist of 104 pubic symphiseal surfaces, but then (line 115) they affirm "[...] The specimens, comprised of 39 males and 19 females [...]". Then, from line 118 to line 120, the sum of the subgroups of the sample again gives 104.

RESPONSE: To facilitate reader understanding, the aforementioned sentence has been revised.

To evaluate the efficacy and accuracy of the forAge program specifically within the Korean population, 104 sides of Korean pubic symphyseal surfaces, donated to the Yonsei University College of Medicine were utilized. These specimens included 39 males and 19 females, aged between 47 to 96 years, with a mean age of 74.5 years, highlighting a focus on middle-aged to elderly demographics. The pubic symphyseal surfaces showing damage from either dissection or bone maceration processes were rigorously excluded. Consequently, the final distribution of the pubic symphyseal sur-face samples analyzed in the study comprised: 10 from specimens in the 40 to 50 age group, 19 from those in their 60s, 38 from those in their 70s, 33 from those in their 80s, and 4 from those in their 90s (Figure 1).

Query 2. Line 161: there is an error in the last part of this line.

RESPONSE: Some words have been modified to make them easier for readers to understand.

The resulting files were exported in the Polygon File Format (.ply) and precisely pre-pared to satisfy the input requirements of the forAge program.

Query 3. From line 167 to line 170: I think this part should be in the next paragraph.

RESPONSE: As your comment, line 167 to 170 moved to the next paragraph.